



# Correcting ozone biases in a global chemistry-climate model: implications for future ozone

Zhenze Liu[1], Ruth M. Doherty[1], Oliver Wild[2], Fiona M. O'Connor[3], and Steven T. Turnock[3,4]

[1]School of GeoSciences, The University of Edinburgh, Edinburgh, UK
[2]Lancaster Environment Centre, Lancaster University, Lancaster, UK
[3]Met Office Hadley Centre, Exeter, UK
[4]University of Leeds Met Office Strategic Research Group, School of Earth and Environment, University of Leeds, Leeds, UK

**Correspondence:** Zhenze Liu (zhenze.liu@ed.ac.uk)

**Abstract.** Weaknesses in process representation in chemistry-climate models lead to biases in simulating surface ozone and to uncertainty in projections of future ozone change. We develop a deep learning model to demonstrate the feasibility of ozone bias correction in a global chemistry-climate model. We apply this approach to identify the key factors causing ozone biases and to correct projections of future surface ozone. Temperature and the related geographic variables latitude and month

show the strongest relationship with ozone biases. This indicates that ozone biases are sensitive to temperature and suggests weaknesses in representation of temperature-sensitive physical or chemical processes. Photolysis rates are also an important factor highlighting the sensitivity of biases to simulated cloud cover and insolation. Atmospheric chemical species such as the hydroxyl radical, nitric acid and peroxyacyl nitrate show strong positive relationships with ozone biases on a regional scale. We correct model projections of future ozone under different climate and emission scenarios following the shared socio-economic

pathways. We find that changes in seasonal ozone mixing ratios from the present day to the future are generally smaller than those simulated without bias correction, especially in high-emission regions. This suggests that the ozone sensitivity to changing emissions and climate may be overestimated with chemistry-climate models. Given the uncertainty in simulating future ozone, we show that deep learning approaches can provide improved assessment of the impacts of climate and emission changes on future air quality, along with valuable information to guide future model development.

## 15  1  Introduction

Atmospheric chemical transport models have been developed over several decades with the principal purpose of simulating the composition of the atmosphere (Zhang, 2008), and chemistry schemes have been incorporated in chemistry-climate and Earth system models to investigate the interactions between atmospheric composition and climate change (Flato, 2011). However, current chemistry-climate models are imperfect in simulating the concentration of atmospheric chemical species, even though

they represent our latest understanding of the governing physical and chemical processes. Biases obtained through comparison with observations indicate that not all relevant processes can be adequately represented in models, and there are uncertainties associated with emissions, chemistry, transport, deposition, clouds, and aerosols in addition to structural errors associated with





model resolution (Knutti and Sedláček, 2013; Archibald et al., 2020a). Representation of these processes may be biased due to poor understanding and simplified parameterisation, and the errors may propagate in complex Earth system models.

While some models reproduce observed concentrations relatively well, this does not confirm that they represent the governing processes well because biases arising from different processes may offset each other. Different models apply differing parameterisations of key processes, and even where these reflect current understanding there may be large differences in model responses to changing conditions (Wild et al., 2020). This may lead to unreliable projections of changes in atmospheric composition under future emission and climate scenarios. However, it is difficult to identify the origin of the biases in models, and

this severely hinders model improvement and prevents a full understanding of the interactions between chemistry and climate through the Earth system.

Tropospheric ozone ($O_3$) is an important greenhouse gas affecting climate, and is a photochemical air pollutant at the Earth's surface, damaging human health and ecosystems (Archibald et al., 2020a). Many studies show that the magnitude of the tropospheric $O_3$ burden and surface $O_3$ concentrations in remote areas can be simulated relatively well (Young et al., 2018;

Griffiths et al., 2021). However, large differences still exist in simulated surface $O_3$ concentrations in high-emission areas (Turnock et al., 2020), and there are large uncertainties in temporal trends (Tarasick et al., 2019) that cannot be captured well by global chemistry-climate models (Parrish et al., 2021). In addition, structural biases in $O_3$ caused by coarse model resolution are hard to eliminate, and typically lead to higher surface $O_3$ concentrations in polluted areas (Wild and Prather, 2006; Stock et al., 2014). Given the difficulty in resolving the $O_3$ biases in a complex chemistry-climate model, the aim of this study is to

correct simulations of present day surface $O_3$ concentrations across the globe, and to generate more reliable $O_3$ projections under future scenarios.

Machine learning provides a valuable approach to correct $O_3$ biases. Appropriate algorithms can be applied to identify the relationships between model responses and the driving variables based on extensive training. Deep learning approaches apply algorithms with more complex architectures and larger parameter spaces based on artificial neural networks (Goodfellow

et al., 2016). In atmospheric science, machine learning has been successfully applied in some fields such as the prediction of precipitation (Sønderby et al., 2020; Ravuri et al., 2021) and air pollution (Kleinert et al., 2021). Numerical approaches used in solving ordinary and partial differential equations in chemical and dynamic systems (Han et al., 2018; Keller and Evans, 2019), and in parameterising subgrid processes for clouds in climate models (Rasp et al., 2018) can also be replaced by machine learning to reduce computational costs. However, reliance on machine learning approaches to make predictions

may lead to loss of interpretability of the results, and we therefore choose to apply a deep learning model to the output from a chemistry-climate model to gain greater physical insight.

In this study, we explore the application of deep learning to correct surface $O_3$ biases in a global chemistry-climate model for the first time, and apply it to improve projections of changes in $O_3$ under future scenarios. We identify the dominant factors leading to $O_3$ biases with the aim of guiding future model development. We introduce the chemistry-climate model, present-

day and future scenarios and the deep learning model in Sect. 2. We demonstrate the performance of the deep learning model in Sect. 3. We show the importance of different variables to $O_3$ biases in Sect. 4, and how these vary by region in Sect. 5.





We quantify surface $O_3$ biases in the present day and future in Sect. 6, and show the importance for assessment of future $O_3$ changes in Sect. 7. We present our conclusions in Sect. 8.

## 2 Approach

### 2.1 Chemistry-climate model and experiments

We use version 1 of the United Kingdom Earth System Model, UKESM1 (Sellar et al., 2019) to simulate present-day (2004–2014) and future (2045–2055) surface $O_3$ mixing ratios under different emission and climate pathways. UKESM1 consists of a physical climate model, the Hadley Centre Global Environment Model version 3 (HadGEM3) with the Global Atmosphere 7.1 and Global Land 7.0 (GA7.1/GL7.0) configurations (Walters et al., 2019) for atmosphere-only simulations with prescribed sea surface temperatures, sea ice and greenhouse gas concentrations generated from the fully coupled UKESM1 (Meinshausen et al., 2017, 2020). Atmospheric composition is modelled with a state-of-the-art chemistry and aerosol module, the United Kingdom Chemistry and Aerosol (UKCA; O'Connor et al., 2014), including a stratosphere-troposphere gas-phase chemistry scheme (StratTrop; Archibald et al., 2020b) and an aerosol scheme (GLOMAP-mode; Mulcahy et al., 2020). An extended chemistry scheme, incorporating more reactive volatile organic compounds (VOCs) is used in this study to provide an improved representation of $O_3$ production environments (Liu et al., 2021). The model resolution is N96L85 in the atmosphere, with 1.875° in longitude by 1.25° in latitude, 85 terrain-following hybrid height layers and a model top at 85 km.

For present day simulations, we use the Coupled-Model Intercomparison Project Phase 6 (CMIP6; Eyring et al., 2016) historical anthropogenic and biomass emissions from Hoesly et al. (2018) and Van Marle et al. (2017) respectively. Biogenic VOC emissions are calculated interactively in the Joint UK Land Environmental Simulator (JULES) land-surface scheme (Pacifico et al., 2011) which is coupled to UKCA. For future simulations, we use the shared socio-economic pathways (SSPs; O'Neill et al., 2014) which represent different pathways of emission and climate policies in the future accounting for social, economic and environmental development (Rao et al., 2017). We choose the SSP3-7.0 and SSP3-7.0-lowNTCF pathways to demonstrate the impacts of weak and strong air pollutant emission controls in the future, respectively. Both pathways lead to a warmer and more humid climate, but SSP3-7.0-lowNTCF has large reductions in anthropogenic emissions of near-term climate forcer (NTCF) species that include $O_3$ precursors and aerosols. Details of the present-day and future emissions under SSP3-7.0 and SSP3-7.0-lowNTCF can be found in Liu et al. (2022). Other emissions used here are the same as described in Turnock et al. (2020).

### 2.2 Deep artificial neural network

We develop a deep learning model using a multilayer perceptron as it is a fundamental approach to build artificial neural networks and easy to apply. More complex approaches such as convolutional or attention-based neural networks could be applied (LeCun et al., 2015; Vaswani et al., 2017), but multilayer perceptron neural networks are competitive and show good





performance compared with other approaches (Tolstikhin et al., 2021). We hence choose a classic artificial neural network as an initial step to explore the possibility of $O_3$ bias correction; more complex approaches could be explored in future.

The multilayer perceptron neural network consists of an input layer, several hidden layers and an output layer, shown in Fig. 1. In the hidden layers, we use three independent modules – a densely-connected layer, a batch-normalisation layer (Ioffe and Szegedy, 2015) and a rectified linear unit (Relu; Glorot et al., 2011). Each layer has neurons that store data and associated weights. Neurons in densely connected layers connect to each neuron in the following layer. The batch-normalisation layers make the model training faster and more stable. The rectified linear unit is a non-linear activation function applied to the output of the previous layer. The deep learning model developed here is applied to correct surface $O_3$ mixing ratios solely simulated by UKESM1.

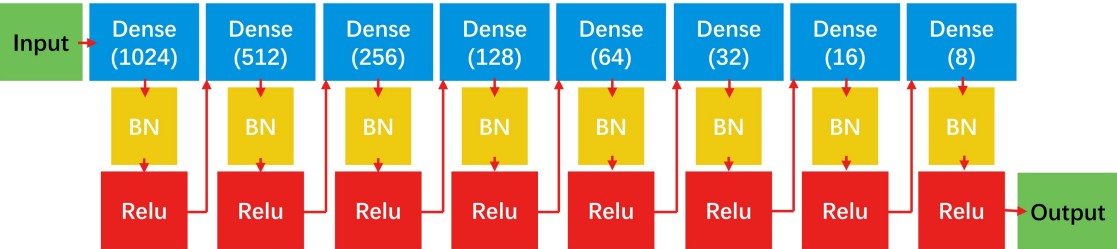

**Figure 1.** The structure of the deep artificial neural network built in this study. Each box represents one layer with neurons and weights to be passed to the next layer. In the densely-connected layer ('Dense') all neurons connect with neurons in the next layer, and the number of neurons is shown in brackets. In the batch-normalisation layer ('BN') the data is normalised and passed to the next layer. The rectified linear unit ('Relu') acts as a non-linear activation function. The arrows show the computation path from input to output.

## 2.3 Deep learning model application

Previous studies have shown that there are systematic seasonal biases in surface $O_3$ mixing ratios simulated with many chemistry-climate models (Young et al., 2018), including UKESM1 (Turnock et al., 2020). We compare present-day UKESM1 results with monthly mean surface $O_3$ reanalysis data from the European Centre for Medium-Range Weather Forecasts (ECMWF) Atmospheric Composition Reanalysis 4 (EAC4) under the Copernicus Atmosphere Monitoring Service (CAMS; Inness et al., 2019). These reanalysis data are generated from chemical assimilation of observations into a chemical transport model, and we choose to use these rather than sparse observations directly as they provide global coverage, which is important for training the deep learning model. The CAMS $O_3$ reanalysis data have been shown to be generally in good agreement with Tropospheric Ozone Assessment Report (TOAR) observations for North America, Europe and parts of Asia where available, although CAMS surface $O_3$ concentrations show some positive biases, particularly in East and Southeast Asia (Huijnen et al., 2020; Wagner et al., 2021). The CAMS reanalysis provides surface concentrations at a scale comparable with our model, and thus avoids uncertainties associated with the spatial representativeness of observations when using measured concentrations





directly. We note that biases in the reanalysis will influence our results, but the CAMS data provide a good foundation with which to demonstrate the feasibility of $O_3$ bias correction.

We find that the mean surface $O_3$ mixing ratios over 2004-2014 simulated by UKESM1 are underestimated in the Northern Hemisphere in winter (December, January, February) and overestimated across most continental areas in summer (June, July, August), as shown in Fig. 2. Surface $O_3$ mixing ratios are overestimated by 4 ppb on an annual mean basis and the biases show a strong seasonal variation in the Northern Hemisphere. The deep learning model is trained to reproduce these biases so that the original UKESM1 $O_3$ mixing ratios can be corrected to match the observation-based reanalysis.

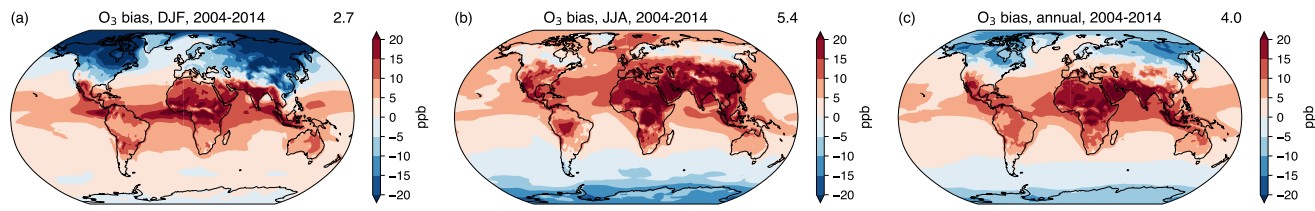

**Figure 2.** Seasonal mean biases in surface $O_3$ mixing ratios simulated with UKESM1 compared with CAMS reanalysis data (UKESM1 minus CAMS) in **(a)** December-January-February (DJF) and **(b)** June-July-August (JJA), and **(c)** annual mean biases, all averaged over 2004–2014. Global area-weighted average surface $O_3$ mixing ratio biases (ppb) are shown in the top right of each panel.

## 115    2.4    Deep learning model input

Earth system models have numerous variables influencing surface $O_3$ mixing ratios, but including all variables as inputs for the deep learning model is impractical due to the heavy computation burden. It may also lead to overfitting, a common issue in machine learning associated with including more variables than can be justified by the limited volume of training data. Limiting the number of variables used as inputs also makes the results easier to interpret. In this exploratory study, we investigated more

than 30 key input variables that represent the major large-scale influences on $O_3$ chemistry and transport, and settled on 20 variables that show the strongest relationships.

We consider major geographical and temporal variables including latitude, longitude, elevation, land cover and month. We define latitude from the equator to the pole, and month from midwinter to midsummer in each hemisphere. Meteorological variables such as temperature, pressure, humidity, zonal and meridional wind are considered as they strongly influence $O_3$

chemical formation and transport. The sensitivity of $O_3$ to temperature is of particular interest, and has been shown to be a substantial source of uncertainty in current studies (Archibald et al., 2020c). Temperature and humidity have also been shown to influence $O_3$ variability on both regional and synoptic scales (Han et al., 2020; Shi et al., 2020). Two fundamental photolysis rates $j(NO_2)$ and $jO(^1D)$ governing $O_3$ production and destruction are considered. Photolysis rates are strongly dependent on clouds, but there are large uncertainties in simulated cloud cover in current models (Wu et al., 2007; Voulgarakis et al., 2009;

Hall et al., 2018). $O_3$ deposition rates and boundary layer height (BLH) are considered as they influence $O_3$ concentrations near the surface (O'Connor et al., 2014; Clifton et al., 2020). Concentrations of $O_3$ precursors such as nitric oxide (NO), VOCs



(primary VOC species) and biogenic isoprene are considered, as these govern $O_3$ chemical production. The concentrations of hydroxyl radical (OH) and the oxidative nitrogen species such as nitric acid ($HNO_3$) and peroxyacyl nitrates (PAN) are also considered because they reflect the general oxidation capacity of the atmosphere. $HNO_3$ and PAN are important nitrogen sinks

that may transport nitrogen and affect $O_3$ formation over a wide area. Between them, the 20 variables selected represent some of the key drivers of uncertainty in simulating surface $O_3$, although we note that they are not independent of each other and that other factors may also be important under some conditions. We use $O_3$ mixing ratios from the lowest model layer of UKESM1 and normalise values of each input variable from zero to one.

## 2.5   Model training

The deep learning model is trained to reproduce the $O_3$ bias in each UKESM1 grid cell based on the corresponding values of the input variables. We train the deep learning model using the biases of monthly mean surface $O_3$ mixing ratios from each model grid cell over 2004–2014 (192 longitudes × 144 latitudes × 12 months × 11 years = 3.6 million data samples). We randomly split the data into training data (80 %), validation data (10 %) and testing data (10 %). Training data are only used to train the model. The validation data provide an evaluation of model performance for each iteration of training, and the testing

data are used to provide an independent evaluation once model training is complete.

The performance of the deep learning model is dependent on the volume of data and the settings used, and we experiment with a range of different settings to keep a balance between training speed and accuracy. We choose an Adam optimiser for the training algorithm (Kingma and Ba, 2014), and use mean absolute error for the loss function in this study. We use 0.01 as the model learning rate, and 1024 grid boxes as the training batch size for stochastic gradient descent. Among these settings, we

find that the batch size is the most important factor influencing the model performance. 1024 randomly sampled data points account for about 4 % of the data from all grid cells in one month in each training iteration, and we find that this is adequate to represent different situations of $O_3$ biases and is found to be sufficient to train the model well.

## 3   Deep learning model performance

We determine the deep learning model performance in predicting surface $O_3$ biases using the testing data to give an independent

evaluation (Fig. 3). The model reproduces the surface $O_3$ biases well with a high correlation coefficient of 0.99 and with a mean bias error of 0.1 ppb and root-mean-square error of 1.9 ppb. The frequency distribution of surface $O_3$ biases predicted by the deep learning model is very similar to that calculated using the $O_3$ reanalysis data. The tails of the distribution also match well, indicating that large biases can be reproduced well. The evaluation demonstrates that the input variables selected are sufficient to predict surface $O_3$ biases well.

To investigate the spatial and temporal behaviour of the model performance, we focus on surface $O_3$ biases in the present-day high-emission regions of North America, Europe, East Asia and South Asia (Fig. 4). North America, Europe and East Asia all show systematic negative surface $O_3$ biases in winter and positive biases in summer (Fig. 4a–c). South Asia shows different behaviour, with consistent positive biases for all months (Fig. 4d). $O_3$ biases in South Asia show more fluctuations over the



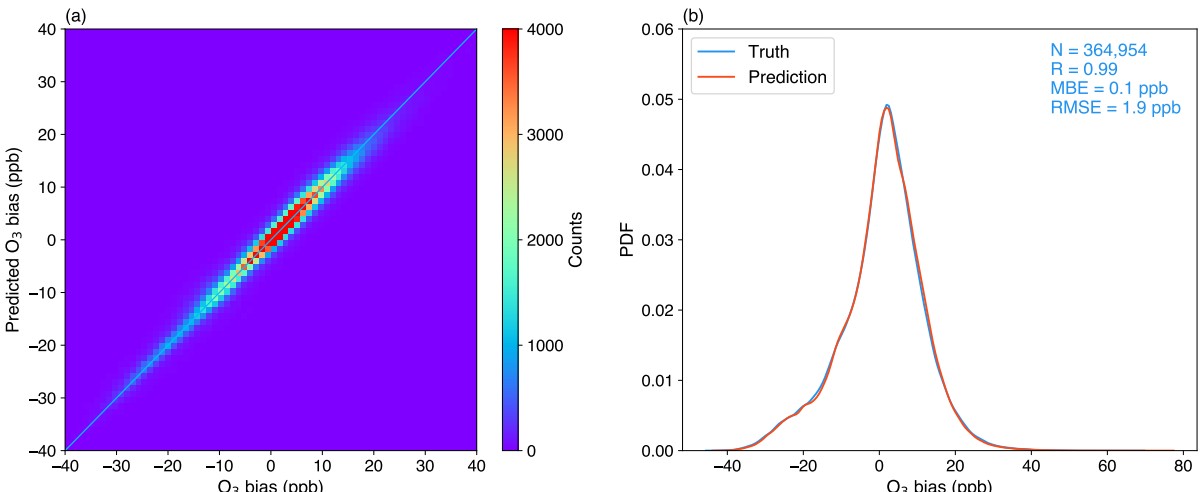

**Figure 3.** Evaluation of the deep learning model in simulating monthly mean surface $O_3$ biases at each UKESM1 grid point based on testing data. **(a)** $O_3$ biases (UKESM1 minus CAMS) and biases predicted by the deep learning model. **(b)** Probability density function of $O_3$ biases (labelled here as Truth) and predicted $O_3$ biases. Statistics are shown in the top right corner.

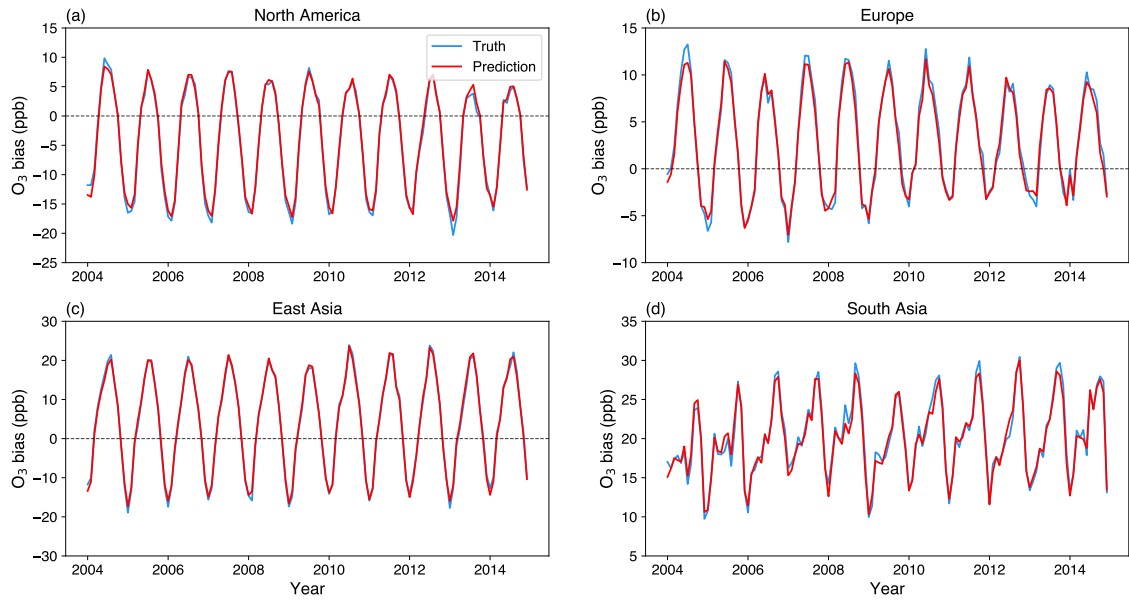

**Figure 4.** Monthly mean surface $O_3$ biases (UKESM1 minus CAMS; Truth) and $O_3$ biases predicted by the deep learning model in **(a)** North America, **(b)** Europe, **(c)** East Asia and **(d)** South Asia from January, 2004 to December, 2014.

annual cycle than those in other regions, but these fluctuations are also captured well by the deep learning model. We note that
the magnitudes of $O_3$ biases are simulated well, and that the differences from year to year are also captured accurately. These






four regions demonstrate that the deep learning model is able to predict regional differences and their respective magnitudes well.

## 4  Feature importance

While all input variables contribute to the prediction of $O_3$ biases, their relative contributions are different and can be estimated
to determine which ones are dominant. An advanced unified framework for interpreting predictions of machine learning models, Shapley additive explanations (SHAP; Lundberg and Lee, 2017) is used to calculate the contribution of different variables to the predicted biases. The feature importance is represented by the SHAP value, which provides a quantitative measure of the variable contribution, shown in Fig. 5. We calculate SHAP values for each variable using 100 sets of 100 data points randomly selected from the full distribution and show their mean values and one standard deviation. The colours indicate the underlying
relationships between the $O_3$ biases and the selected variables based on the correlation between the calculated SHAP values and variable values. Red represents a strong positive relationship ($r > 0.7$), blue represents a strong negative relationship ($r < -0.7$), and green shows weaker relationships.

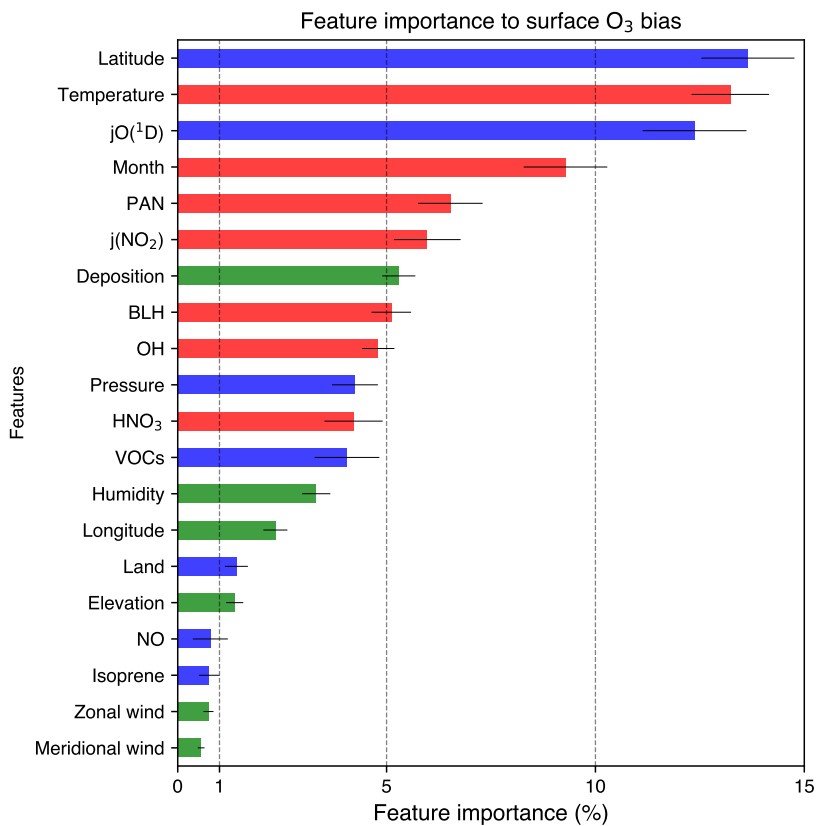

**Figure 5.** Importance of different variables to surface $O_3$ biases calculated by the Shapley additive explanations framework (SHAP) for the deep learning model. Strong positive (r > 0.7) and negative (r < -0.7) relationships between $O_3$ biases and variable values are shown in red and blue, respectively, while weaker relationships are shown in green. The error bars show one standard deviation of feature importance for each variable.

We find that latitude and month are important to $O_3$ biases, and show negative and positive relationships to surface $O_3$ biases, respectively. This reflects more positive biases in Tropical regions than at the Poles, and more positive biases in summer than winter. Temperature also shows a strong positive relationship, and this may partly reinforce the influence of latitude and month. Photolysis rates are also important for $O_3$ biases, with $jO(^1D)$ associated with $O_3$ destruction and $j(NO_2)$ with $O_3$ production. The concentrations of PAN, OH and $HNO_3$ all show positive relationships to $O_3$ biases. This may indicate that there are large uncertainties in $O_3$ production under high-oxidation and high-$NO_x$ environments. However, we find that VOCs and short-lived NO concentrations are less important to $O_3$ biases. This highlights the systematic regional and global-scale nature of the $O_3$ biases in UKESM1, and indicates that the biases are not strongly associated with precursor abundance on a regional level. Similarly, isoprene concentrations show little contribution to $O_3$ biases. We note that while $O_3$ deposition rates and BLH are





both important to O$_3$ biases, this may partly reflect their similar seasonality. We note that the relationships derived between the variables and O$_3$ biases reflect association, not direct causation.

The relationships between variables with highest feature importance and the O$_3$ biases are generally directly interpretable,
demonstrating that the deep learning model may be capturing the internal relationships between inputs and outputs in a physically realistic way. This provides some insight into the sources of O$_3$ biases in UKESM1. We emphasise that the high importance of a variable does not indicate that the variable itself is not simulated well by the chemistry-climate model, or that it is the direct cause of the bias. Since temperature is generally represented well in UKESM1 (Sellar et al., 2019), the importance of temperature thus indicates that O$_3$ biases may be caused by the representation of physical and chemical processes that are
sensitive to temperature changes, such as chemical reaction rates (Coates et al., 2016; Newsome and Evans, 2017), or to other processes for which temperature is a proxy, and this explains the seasonality of the reversal in O$_3$ biases from winter to summer in the Northern hemisphere.

## 5  Spatial O$_3$ bias sensitivity

The sensitivity of surface O$_3$ biases to specific variables differs across regions, and we show the spatial sensitivity to variables
with high feature importance and strong correlation to O$_3$ biases in Fig. 6. Since each variable is considered independent in the deep learning model, we use the change in annual mean O$_3$ bias caused by changes in each variable in each UKESM1 grid cell independently to represent the spatial sensitivity. We perform an experiment for each variable where we increase the value of that variable by a small amount (0.5 standard derivations of its temporal variability over 2004-2014) and calculate the corresponding change in surface O$_3$.

Surface O$_3$ biases are most sensitive to temperature, particularly in continental areas in the Northern hemisphere where higher temperatures are associated with higher O$_3$ (Fig. 6a). There is a strong relationship with photolysis rates across a large area, particularly in continental areas at mid and high latitudes (Fig. 6b, c), and there is a larger influence from jO($^1$D) than from j(NO$_2$). The chemical environment is important for O$_3$ biases on a regional scale. OH concentrations show a strong association with O$_3$ biases in North America, Europe and East Asia, indicating that high biases in these high-emission regions may be
associated with high atmospheric oxidation capacity (Fig. 6d). There is also a strong sensitivity to the concentrations of PAN in South Africa, South Asia and South East Asia (Fig. 6e). This may indicate uncertainty in the NO$_x$ emission inventory in these regions or the large impacts of nitrogen reservoirs on O$_3$ production. Given the long lifetime of PAN, it is also associated with O$_3$ biases in remote areas such as the Arctic, indicating that the transport of air pollutants may be important to surface O$_3$ in these areas. BLH is associated with O$_3$ biases in tropical oceanic areas (Fig. 6f), and this may reveal the importance of greater
O$_3$ mixing and downward transport when the boundary layer is relatively deep.

The spatial sensitivity of surface O$_3$ biases to different variables is helpful to guide future improvement of the UKESM1 model. There are substantial changes in annual mean surface O$_3$ biases associated with adjusting variables values. Increasing temperature, jO($^1$D), j(NO$_2$), OH and PAN concentrations by 0.5 standard deviation changes annual mean surface O$_3$ biases from 4.0 ppb to 4.8 ppb (20 %), 3.0 ppb (-25 %), 4.3 ppb (8 %), 4.5 ppb (13 %) and 4.7 ppb (18 %), respectively. However,



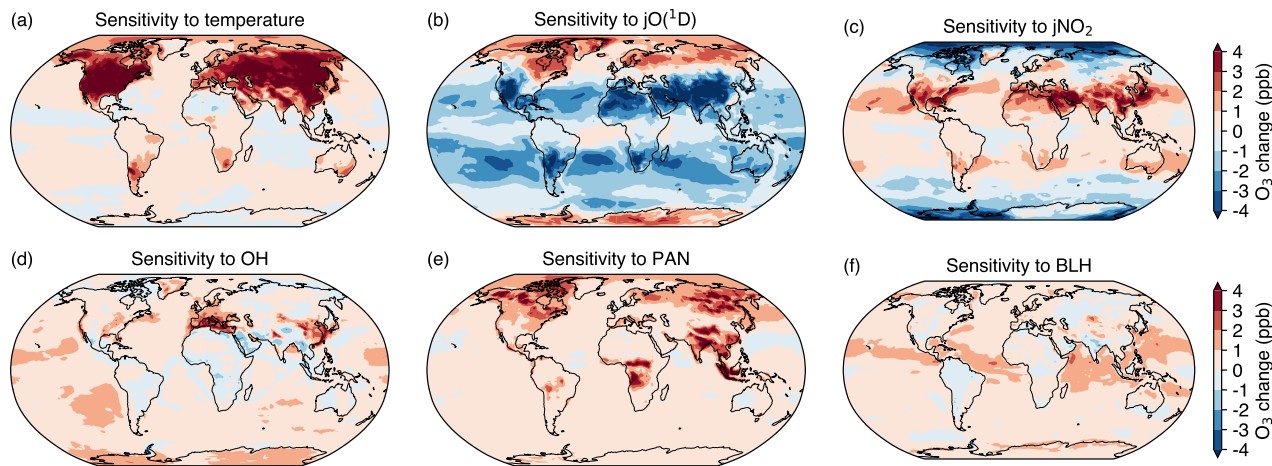

**Figure 6.** Sensitivity of annual mean surface $O_3$ bias to increases in **(a)** temperature, **(b)** $jO(^1D)$, **(c)** $j(NO_2)$, **(d)** OH, **(e)** PAN and **(f)** BLH. Variable values are increased by 0.5 standard deviation of their temporal variability for each UKESM1 grid cell independently.

we note that UKESM1 generally reproduces temperature and photolysis rates well compared with observations (Telford et al., 2013; Sellar et al., 2019), although there are large differences in simulated concentrations of OH and PAN (O'Connor et al., 2014; Nicely et al., 2020). Our results suggest that chemical processes associated with temperature and oxidation capacity, and cloud and aerosols influencing photolysis rates may be important sources of $O_3$ biases in UKESM1, and that improved representation of these processes may reduce current biases in surface $O_3$.

## 6  Assessing biases in modelled future surface $O_3$

We can apply the relationships between variables and surface $O_3$ biases derived from present day simulations to assess the biases in future $O_3$ projections with UKESM1 and to correct our estimates of future $O_3$ concentrations. We demonstrate how surface $O_3$ biases change for two future emission and climate scenarios, SSP3-7.0 and SSP3-7.0-lowNTCF. These pathways are associated with a warmer and more humid climate than the present day. While increased temperature might be expected to increase surface $O_3$ biases, we find that annual mean $O_3$ biases decrease from 4.0 ppb to 3.6 ppb (11 %) under SSP3-7.0 and to 1.3 ppb (67 %) under SSP3-7.0-lowNTCF. This is principally due to the changes in the chemical environment reflected by decreases in the concentrations of OH (-15 % and -13 %) and PAN (-30 % and -38 %) under SSP3-7.0 and SSP3-7.0-lowNTCF, respectively. In continental areas where surface $O_3$ concentrations are overestimated, the UKESM1 model performance is likely to improve under these less polluted future conditions. Since SSP3-7.0-lowNTCF represents a more stringent emission control pathway than SSP3-7.0, there are larger decreases in $O_3$ biases under this scenario.




We investigate the spatial distribution of annual mean changes in surface $O_3$ biases in future scenarios. We find that $O_3$ biases decrease in most oceanic areas under both future scenarios, see Fig. 7. However, $O_3$ biases increase in some continental areas especially in the Middle East, South Asia and East Asia under SSP3-7.0. This is due to less stringent emission controls in these regions and hence higher concentrations of $O_3$ precursors and their oxidation products under SSP3-7.0 (Turnock et al.,

2020). Under SSP3-7.0-lowNTCF, there are widespread decreases in $O_3$ biases except over East Asia, where anthropogenic VOC emissions increase substantially and there is a corresponding increase in PAN concentrations and an increase in $O_3$ biases. In high-emission regions, the performance of UKESM1 in future $O_3$ simulations largely depends on changes in $O_3$ precursor emissions given that changes in temperature and photolysis rates are small under future scenarios. The performance of UKESM1 in high-emission regions is expected to improve under scenarios with clean air quality policies, but is likely to

become worse under scenarios with increasing future pollutant emissions.

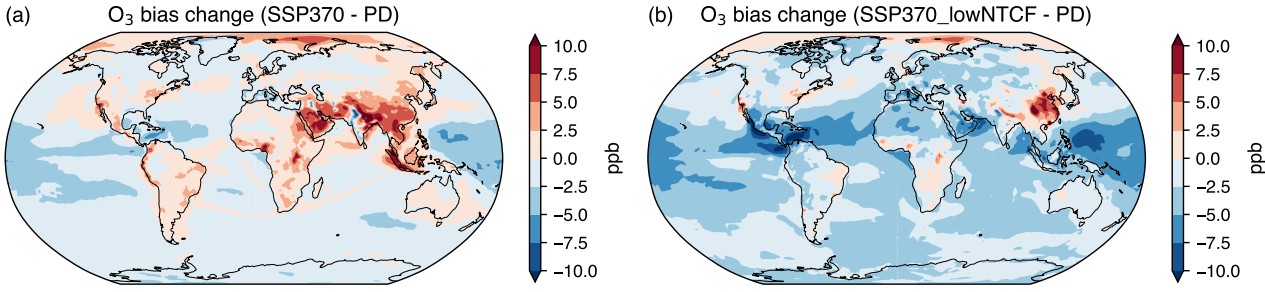

**Figure 7.** Annual mean change in surface $O_3$ biases (ppb) between the present day (PD) and 2045-2055 under **(a)** SSP3-7.0 and **(b)** SSP3-7.0-lowNTCF pathways.

## 7 Bias correction in future $O_3$ projections

We can provide more reliable projections of future $O_3$ by subtracting the calculated surface $O_3$ biases from surface $O_3$ mixing ratios simulated with UKESM1 under future scenarios (Fig. 8). The simulated surface $O_3$ mixing ratios vary in the different scenarios due to different emissions and climate (Fig. 8a-c), but the spatial distributions are generally similar, with the highest

$O_3$ levels in the Middle East and South Asia. The spatial patterns of surface $O_3$ biases are also similar under the different scenarios, with biases highest in the Tropics (Fig. 8d-f). High $O_3$ mixing ratios in the Middle East and South Asia are reduced greatly after $O_3$ bias correction (Fig. 8g-h). There are also large decreases in surface $O_3$ mixing ratios in high-emission regions e.g. North America and East Asia, and continental outflow regions e.g. North Atlantic. The corrected global annual mean surface $O_3$ mixing ratios are lower than those simulated under all scenarios, and are highest under SSP3-7.0 and lowest under

SSP3-7.0-lowNTCF, which is consistent with the uncorrected UKESM1 results.

We show the changes in seasonal mean surface $O_3$ mixing ratios in North America, Europe, South Asia, East Asia and the globe from the present day to the future in Fig. 9, comparing the original assessments using UKESM1 with the bias-corrected



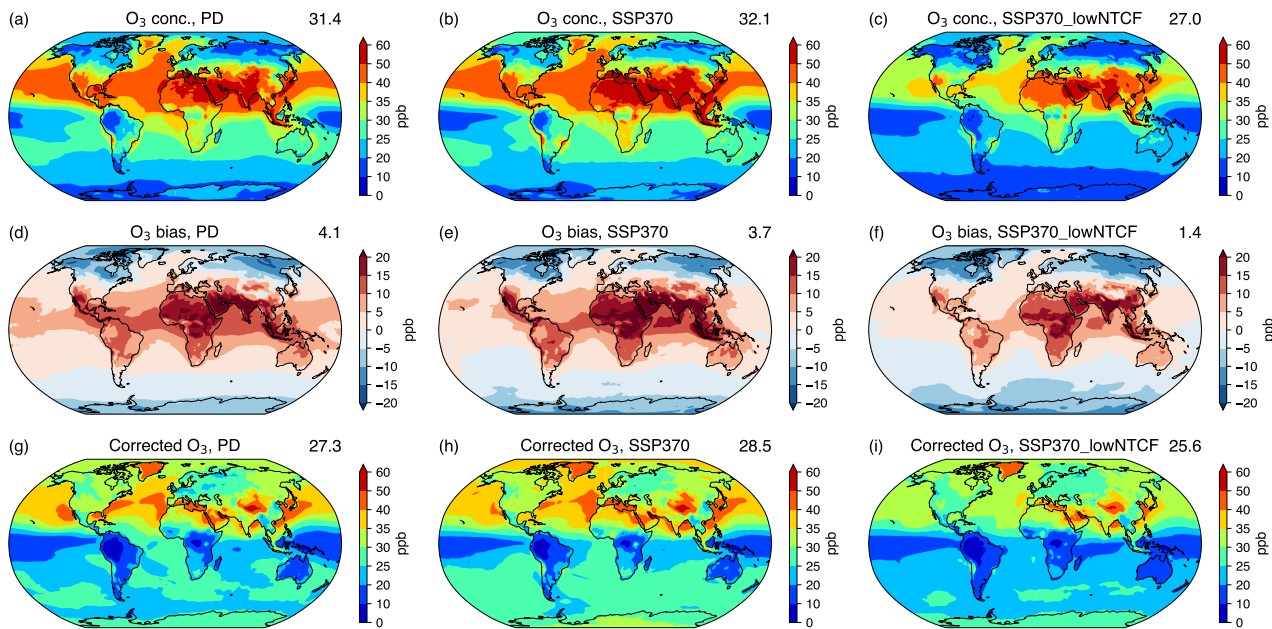

**Figure 8.** Annual mean surface $O_3$ mixing ratios (ppb) from UKESM1 simulations for **(a)** the present day (PD), **(b)** SSP3-7.0 and **(c)** SSP3-7.0-lowNTCF. The corresponding surface $O_3$ biases predicted with the deep learning model are shown in **(d-f)** and corrected surface $O_3$ mixing ratios are shown in **(g-i)**. Annual global mean mixing ratios are shown in the top right of each panel.

values. Under SSP3-7.0, the corrected changes in global mean surface $O_3$ are slightly larger than the uncorrected UKESM1 results. However, in high-emissions regions the corrected changes are generally smaller than those originally simulated under

both SSP3-7.0 and SSP3-7.0-lowNTCF. In summer, corrected surface $O_3$ mixing ratios increase in all regions considered here under SSP3-7.0, and decrease under SSP3-7.0-lowNTCF. Corrected $O_3$ increases in South and East Asia under SSP3-7.0 are 6-8 ppb smaller than those simulated, and this indicates that $O_3$ air quality degradation due to future emission growth and climate change may not be as severe as the uncorrected UKESM1 simulations suggest. Similarly, under SSP3-7.0-lowNTCF, corrected $O_3$ decreases are smaller in all regions, and this indicates that the impacts of emission controls on $O_3$ mitigation may

be smaller than those expected. This can be confirmed by the smaller global mean $O_3$ decreases under SSP3-7.0-lowNTCF in the bias-corrected assessment ($< 2$ ppb) than in the original UKESM1 simulation ($> 3$ ppb). In winter, the corrected changes in surface $O_3$ mixing ratios are smaller than those simulated with UKESM1, whether these changes are positive or negative.

These results highlight that the influence of changing emissions and climate on $O_3$ may not be as large as those simulated with UKESM1 and thus projections of future surface $O_3$ changes may be overestimated. UKESM1 shows a strong seasonality

in surface $O_3$ likely due to strong $O_3$ sensitivity to temperature and chemical environment, and this leads to large changes in future $O_3$. UKESM1 typically overestimates future surface $O_3$ changes, and other chemistry-climate models are likely to display similar behaviour. Therefore, the impacts of changes in emissions and climate on future $O_3$ should be re-assessed in light of the underlying surface $O_3$ biases. We demonstrate the successful application of a deep learning model to address this



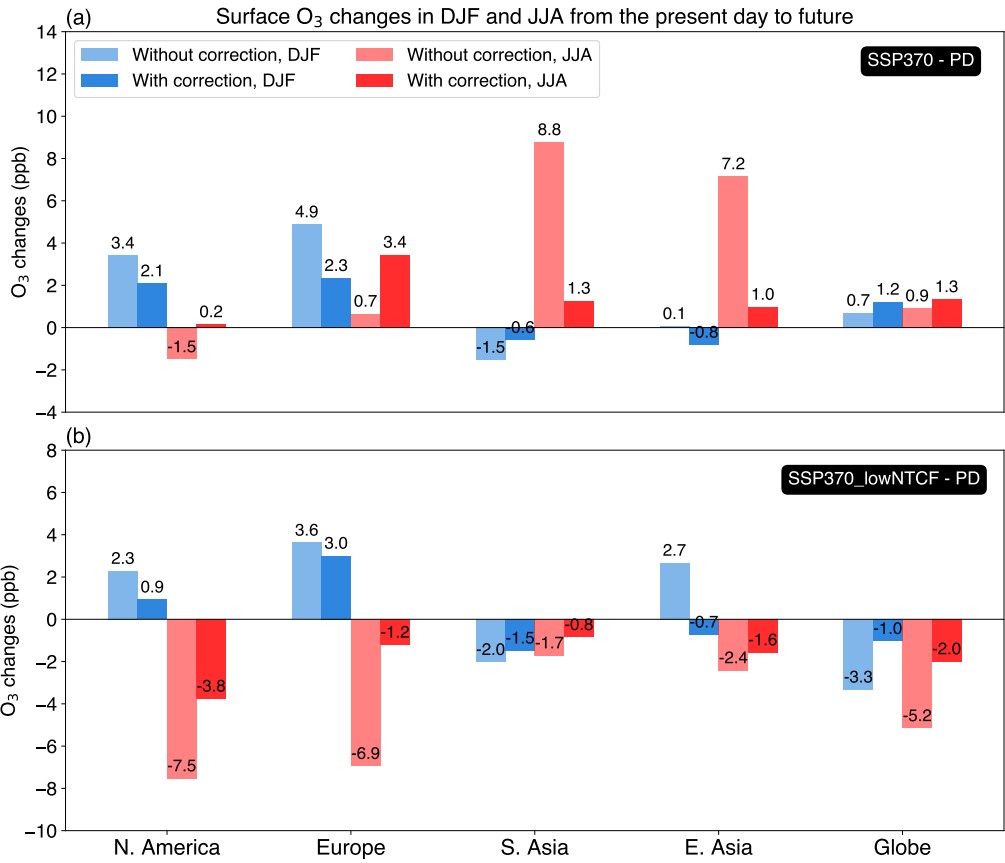

**Figure 9.** Changes in seasonal mean surface $O_3$ mixing ratios (ppb) with and without corrections in DJF (blue bars) and JJA (red bars) from the present day (PD) to **(a)** SSP3-7.0 and **(b)** SSP3-7.0-lowNTCF in North America, Europe, South Asia, East Asia and the globe.

issue, and it would be valuable to take a similar approach with the output of other chemistry-climate models to provide a more reliable assessment of future surface $O_3$ changes.

## 8 Conclusions

There are large uncertainties in the simulation of surface $O_3$ in current chemistry-climate models, but it is difficult to identify the causes of biases and improve representation of the key processes. In this study, we have demonstrated the feasibility of correcting surface $O_3$ biases for a chemistry-climate model, UKESM1, using a machine learning technique. A deep artificial neural network is built with input variables important for $O_3$ chemistry and dynamics. The deep learning model shows good performance in predicting surface $O_3$ biases, with a high correlation coefficient of 0.99 and small mean bias errors of 0.1 ppb.





Application of the deep learning model to the results from the process-based UKESM1 model shows promise for predicting future $O_3$ concentrations under different climate and emission trajectories with greater confidence.

This study has also explored the key factors governing $O_3$ biases, which provide valuable insight for model improvement. We find that temperature is an important factor governing $O_3$ biases, especially for continental areas in the Northern hemisphere, indicating that physical and chemical processes influenced by temperature may be not represented well. Photolysis rates also contribute to $O_3$ biases across the globe, indicating that simulated clouds and aerosols may be an important source of $O_3$ biases. Chemical species such as PAN and OH are closely associated with $O_3$ biases on a regional scale, suggesting that weaknesses in representation of key chemical processes remains a substantial issue.

We have applied a deep learning model to generate a correction to the projections of surface $O_3$ mixing ratios for the present day and under future SSP3-7.0 and SSP3-7.0-lowNTCF pathways. We find that global annual mean $O_3$ biases (4.0 ppb) decrease by 0.4 ppb (11 %) and 2.7 ppb (67 %) under these scenarios, respectively. However, $O_3$ biases in high-emissions areas may increase due to increased $O_3$ precursors. We use this approach to demonstrate that seasonal changes in surface $O_3$ mixing ratios from the present day to the future may be overestimated by as much as 6 ppb with UKESM1, especially in high-emission areas, and this highlights a strong $O_3$ sensitivity to changes in future emissions and climate in the model. A similar overestimation of future $O_3$ changes is likely in other chemistry-climate models, and the influence of emission controls on surface $O_3$ mixing ratios may thus be smaller than suggested by current model simulations. This suggests that emission control policies may be less effective in improving regional air quality than global model simulations indicate.

The deep learning model employed here is a valuable tool to obtain more reliable predictions of the magnitude and spatial distribution of surface $O_3$ mixing ratios. We acknowledge that the choice of input variables is likely to influence the sensitivity of $O_3$ biases derived from the deep learning model, and the relationships between $O_3$ biases and input variables are not always readily interpretable, which is common in machine learning. However, we demonstrate that the relationships between the variables with the highest feature importance and surface $O_3$ biases are intuitive, e.g. with temperature and photolysis rates, and this provides useful insight for further model improvement. We acknowledge that there are uncertainties in the use of reanalysis data, but have successfully demonstrated the feasibility of bias correction using this data, and will explore the challenges of data sparsity and spatial representativeness associated with use of surface measurements directly in future work. The approach applied here provides a valuable opportunity to examine the uncertainties in a chemistry-climate model, and helps improve assessment of the impacts of changing emissions and climate on future air quality.

*Data availability.* The data generated in this study are available upon request.

*Author contributions.* ZL, RD, OW designed the study. ZL built the model, conducted model simulations and performed the analysis with input from OW, RD, FO'C and ST. ZL, RD and OW prepared the paper, with contributions from all co-authors.



*Competing interests.* The authors declare that they have no conflict of interest.

*Acknowledgements.* Zhenze Liu thanks the University of Edinburgh China Scholarship Council. Oliver Wild and Ruth M. Doherty thank the Natural Environment Research Council (NERC) for funding under grants NE/N006925/1, NE/N006976/1 and NE/N006941/1. Fiona M.
315 O'Connor was supported by the Met Office Hadley Centre Climate Programme funded by BEIS and also acknowledges support from the EU Horizon 2020 Research Programme CRESCENDO (grant agreement number 641816). Steven Turnock would like to acknowledge support from the UK-China Research and Innovation Partnership Fund through the Met Office Climate Science for Service Partnership (CSSP) China as part of the Newton Fund.



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
