# Peer review of "Correcting ozone biases in a global chemistry-climate model: implications for future ozone"

_Atmospheric Chemistry and Physics, 2022_

## Author Comment (AC1)

Dear editor and all reviewers:

We thank the editor and reviewers for their contribution to the improvement of our manuscript. Responses to reviewers on "Correcting ozone biases in a global chemistry-climate model: implications for future ozone" by Zhenze Liu et al. are given below. For clarity, the reviewer comments are given in bold, followed by our responses. The modified text in our revised manuscript is given in italics and blue.

**Reviewer comments 1:**

**This work addresses the uncertainties in the simulation of surface ozone in chemistry-climate models by offering a novel approach, using machine learning, in identifying the causes of biases originating from the representation of related key processes. As such, it is an excellent feasibility contribution in correcting modelled $O_3$ biases. The results demonstrate that these biases are mainly due to the UKESM1 model effect of temperature and photolysis rates, and thus provides valuable insight for further model improvement which ultimately should allow improving the assessment of the impacts of changing emissions and climate on future air quality. The paper is very well written and supported with solid analysis, clear presentation and meaningful interpretation and I have no comments of technical nature. I would only like to point out the need for follow-up studies to tackle the questions of the sensitivity of the obtained results on the particular machine learning technique employed here and the environment of the technique validation with the consideration of additional, independent derivations of the $O_3$ biases with alternative observational data. These are included in the following specific comments indicated by the corresponding line numbers.**

We thank the reviewer for their positive comments here, and address specific comments below.

**SPECIFIC COMMENTS**

1. **49-51: This sentence does not clearly introduce the motivation/justification of this study. How is the "gain of greater physical insight" consistent to "loss of interpretability of the results"? Please rephrase.**

The motivation of this study is to explore the feasibility of correcting modelled surface ozone biases, and to provide improved future ozone projection. However, the approach we use also allows us to extract information on the relationship between key model variables and model biases. While the approach does not allow us to attribute the biases to specific processes in the model directly, it does provide valuable insight into the origin of the biases in a physically meaningful way. The biases are strongly related to temperature, for example, which implicates chemical and/or boundary layer processes. This provides some guidance for future model development, and goes beyond a simple black-box prediction of future biases, which is what our phrase "loss of interpretability" was referring to. We thank for reviewer for their comment and we have now rephrased this sentence to

make the meaning clearer.

Page 2, line 50:
*However, reliance on machine learning approaches to make predictions may lead to loss of interpretability of the results.* *We therefore choose an approach based on physical model variables that allows us to extract the importance of these variables and thus derive some physical insight into the performance of the chemistry-climate model.*

**2. 52-53: Is the current study the first one to apply machine learning for estimating and correcting global model ozone biases? If yes it should be mentioned explicitly.**

As far as we are aware, our study is the first to demonstrate ozone bias correction for a global chemistry-climate model with the purpose of producing improved projections of future ozone. Two other studies have corrected ozone biases in chemistry transport models, but they have focused only on matching present-day ozone concentrations (Ivatt and Evans, 2020; Keller et al., 2021). We have amended the text to make the originality clearer.

Page 2, line 54:
*In this study, we explore the application of deep learning to correct surface $O_3$ biases in a global chemistry-climate model, and we apply it for the first time to improve projections of changes in $O_3$ under future scenarios.*

**3. 102-103: Can this statement be elaborated? Is there no value in repeating the process (in a separate study perhaps) including observed station data from locations around the globe as to verify the results from the current gridded data analysis?**

We chose CAMS reanalysis data as they match observed surface ozone much better than UKESM1, and because they provide global data coverage. It is valuable to have a large amount of data to train machine learning models to ensure robust results. However, we acknowledge that CAMS reanalysis data does not always match observations where these are available, and note that this has been explored in more detail in other studies (Huijnen et al., 2020). Observations could be used directly, but we caution that these also have substantial uncertainties associated with spatial representativeness and data sparsity. We have now introduced the TOAR dataset to provide a more comprehensive comparison between different surface ozone datasets in Figure 2 of the revised manuscript. We have placed Section 'Deep learning model application' after Section 'Deep learning model input', and added new discussion text on TOAR and other observation-based dataset and their limitations.

Page 6, line 144:
*2.4 Deep learning model application*
*Previous studies have shown that there are systematic seasonal biases in surface $O_3$ mixing ratios simulated with many chemistry-climate models (Young et al., 2018), including UKESM1 (Turnock*

*et al., 2020). Ozone observations, such as those compiled for the Tropospheric Ozone Assessment Report (TOAR; Schultz et al., 2017), are typically used to evaluate model performance but observation sites are sparsely distributed, and there are few outside North America, Europe and parts of East Asia. In addition, many observations are representative of much smaller spatial scales than can be resolved by coarse resolution models, and this presents an additional source of uncertainty.*

*We therefore also consider surface O₃ reanalysis data from the European Centre for Medium-Range Weather Forecasts (ECMWF) Atmospheric Composition Reanalysis 4 (EAC4) under the Copernicus Atmosphere Monitoring Service (CAMS; Inness et al., 2019). This data is at a similar spatial scale to UKESM1 output and provides global data coverage, which is valuable in training the deep learning model to ensure more robust results. We compare surface O₃ in UKESM1 with TOAR observations and CAMS reanalysis in Fig. 2. There are substantial biases in UKESM1, with surface O₃ underestimated compared with observations in wintertime and overestimated in summertime. In contrast, the CAMS reanalysis is in much better agreement with TOAR, with mean seasonal biases of about 3 ppb. Comparing UKESM1 and CAMS data over the globe, we find that mean surface O₃ mixing ratios over 2004-2014 simulated by UKESM1 are underestimated in the Northern Hemisphere in winter (December, January, February) and overestimated across most continental areas in summer (June, July, August), and this occurs over broad regions, not just where observations are available.*

*In the absence of a global observation-based ozone climatology, we apply the CAMS reanalysis product in our analysis. We note that recent studies have explored the fusion of observations and model output to generate surface O₃ products at a global scale (Chang et al., 2019; Betancourt et al., 2022), but these approaches only work well in regions where measurement sites are available. The CAMS reanalysis provides surface concentrations at a scale comparable with our model, and thus avoids uncertainties associated with the spatial representativeness of observations when using measured concentrations. While biases in the reanalysis will influence our results, the CAMS data provide a good foundation with which to demonstrate the feasibility of O₃ bias correction.*

[Figure]

**Figure 2.** Comparison of seasonal mean (December-January-February (DJF) and June-July-August (JJA)) and annual mean surface $O_3$ mixing ratios between **(a-c)** UKESM1 and TOAR, **(d-f)** CAMS and TOAR, and **(g-i)** UKEMS1 and TOAR, all averaged over 2004-2014. Global area-weighted average surface mean mixing ratios (ppb) are shown in the top right of each panel.

4. **151-154: In relation to the previous comment and in conjuction to the method description in the first paragraph of sub-section 2.5, the results of this test are impressive but of course must be viewed in the framework of comparing a test run against a validation run of the same (machine leaning) model. Is it possible to compare the resulting O3 biases from this test run (for selected locations) with respective biases obtained from a conventional method and an alternative observational data source?**

The independent evaluation shown in Figure 3 demonstrates that the deep learning model performs very well in reproducing the UKESM1 bias at times and locations that it was not trained at. However, the testing data provide sparse coverage over all times and locations, and thus sampling at a particular location would provide only partial (10%) coverage. The predicted bias would match the bias in UKESM1 with respect to CAMS very well (as evident from Fig 3), but any biases with respect to independent observations would then just reflect differences between CAMS and those observations, which provides little additional insight. However, we now include the biases in CAMS compared with TOAR observations in the new version of Figure 2, as discussed in detail above, to provide clearer context.

5. **186-188: Can the results and the conclusions of Fig.5 be compared to those of other studies (based on any other methodology)?**

As noted, a few studies have used machine learning to calculate feature importance for biases in ozone (Ivatt and Evans, 2020; Keller et al., 2021). The most important variables these studies identify are broadly similar to those that we find, e.g. time of year and precursors such as PAN, but others differ, reflecting the different focus of the studies and different choice of features investigated, as well as differences in the models and approaches used. We note that these studies both used far more variables as inputs, and this presents a greater risk of overfitting which would also be expected to influence feature importance. However, we have now included a sentence in the paper to compare the feature importance with those of these other studies.

Page 10, line 217:

*… We note that while $O_3$ deposition rates and BLH are both important to $O_3$ biases, this may partly reflect their similar seasonality. Previous studies investigating model $O_3$ biases have found a broadly similar importance for some variables, e.g. for time of year and precursors such as PAN, but the different focus of these studies make direct comparison of the results difficult (Ivatt and Evans, 2020; Keller et al., 2021).*

6. **296-298: Also the choice of the particular machine learning model influences the derived O3 biases and conclusions.**

We thank the reviewer for this point, and we have added it into text.

Page 17, line 353:
*We acknowledge that the choice of input variables and the machine learning approach applied are both likely to influence the sensitivity of $O_3$ biases derived from the deep learning model ...*

**Reviewer comments 2:**

**Overview:**

**The paper presents a multilayer perceptron ML approach to predict the differences between monthly-mean simulations of ozone surface concentrations simulated by the UKESM1 and the EAC4 CAMS Reanalysis. The ML approach manages to predict the differences to a good degree, which is exploited by the authors to ML predict the differences for UKESMI1 scenarios simulation of future ozone surface concentrations. The ML approach is trained by set of 20-30 input variables, such as temperature, photolysis rates and boundary layer height. While not a direct outcome of the ML approach, the importance of the different input variables is quantified using the SHAP framework. The paper is well written.**

We thank the reviewer for their positive comments here, and address the two main concerns below.

**General review:**
1. **The fundamental problem of the paper is the use of the ozone surface concentration fields of the CAMSRA data as the "truth". The ozone surface analysis is corrected too little by the assimilated ozone satellite retrievals, which are dominated by the stratospheric signal, to be considered a representation of the "truth" or the observations; they are just another model run. https://atmosphere.copernicus.eu/sites/default/files/2021-06/CAMS84_2018SC3_D5.1.1-2020_reanalysis_validation.pdf. CAMS84_2018SC3_D5.1.1-2020_reanalysis_validation (copernicus.eu). There is no continuous gridded data set, which would make the identification of biases of simulated ozone surface values for all regions of the earth surface a straightforward task. This is in contrast to meteorological simulations for which meteorological re-analysis (i.e. ERA5) are considered a continuous global reference data set.**

We acknowledge the reviewer's concern about our use of CAMS reanalysis data, and considered these issues carefully when originally selecting what surface ozone data to apply. We are aware that surface ozone observations are not used in the assimilation system, but note that assimilation of

ozone elsewhere in the atmosphere leads to better representation of surface ozone than is available from other unconstrained model products. Direct comparison of CAMS surface ozone with observations demonstrates reasonably good performance, despite these limitations (Huijnen et al., 2020), and the remaining biases are far smaller than those present in the UKESM1 model that we are using. We highlight that the purpose of our study is to demonstrate a new approach to bias correction, and this is independent of the choice of ozone data used. Direct use of observational data has its own caveats associated with the representativeness of measurement locations of the larger spatial scales that we resolve in our model (~150 km scale) and with sparse data coverage, and we note that these are very likely the same reasons that surface observations are not assimilated in the CAMS reanalysis.

We agree with the reviewer that there is no perfect surface ozone climatology yet available to apply, but we do not feel that this compromises the value of our study. We have chosen to use the CAMS reanalysis data because it provides a reliable and realistic surface ozone climatology at scales comparable with UKESM1 that has complete coverage in space and time across the world. However, to address the reviewer's concern, we have now explored the use of TOAR surface observations directly, and have included this in Section 2.4 'Deep learning model application' as detailed above in our response to reviewer 1.

On a separate point, we note that the word 'Truth' is commonly used in the machine learning field to represent training targets. However, we appreciate that it may be considered misleading in the current context, and we have therefore replaced this word with 'Reference' in the text and figures throughout the paper.

2. **For this reason, I suggest rejecting the paper because I cannot think of an easy fix of that dilemma. It seems necessary to introduce independent surface in-situ observations such as the TOAR data sets. Using the TOAR data set as reference make sense but limits strongly the area for which biases can be determined. The TOAR data set could also be used to demonstrate that the biases of UKSEM1 are indeed much larger than the biases of CAMS RA. An alternative approach could be to use the applied method to transition results between different model types, but I am not sure if there are useful applications for that.**

We appreciate the reviewer's concern here, but we do not feel that lack of an ideal surface ozone climatology with which to compare invalidates our study. Our demonstration of the approach, and our finding that ozone sensitivity to future climate and chemistry are overestimated in present models are still both valid. Future development of a robust and reliable ozone climatology would be valuable and would allow us to provide an improved quantification of the magnitude of the biases, but the main conclusions of our paper remain unaffected.

However, we agree that it would be valuable to explore the use of surface ozone observations, despite the limitations associated with sparse coverage and spatial representativeness. We have therefore applied the TOAR surface ozone dataset, as suggested, and compared it with CAMS

reanalysis data. We note that differences between CAMS and TOAR are much smaller than those between UKESM1 and TOAR, highlighting the large room for improvement of UKESM1, and we now show these differences in Figure 2 and accompanying text, as discussed in our responses to reviewer 1 and in Figure 5. We have also compared the feature importance that we derive for surface ozone biases based on CAMS and TOAR data separately, and describe these in Section 4. We find that the most important variables inferred from both datasets are the same, and this confirms that use of CAMS data does not fundamentally impact our results. However, in acknowledgement of the reviewer's concerns we have now expressed our outlook for future development of the technique in the Conclusion section.

Page 10, line 217:

*...We note that while $O_3$ deposition rates and BLH are both important to $O_3$ biases, this may partly reflect their similar seasonality. Previous studies investigating model $O_3$ biases have found a broadly similar importance for some variables, e.g. for time of year and ozone precursors such as PAN, but the different focus of these studies make direct comparison of the results difficult (Ivatt and Evans, 2020; Keller et al., 2021).*

*To highlight the sensitivity of our results to the physical and chemical environment, we show the feature importance over land and ocean regions separately in Fig. 5b. Ozone precursors such as NO, isoprene and VOCs are much more important over land, along with some physical variables such as temperature. In contrast, $O_3$ biases over the ocean are more sensitive to OH, deposition and boundary layer mixing. These differences reflect the differing importance of $O_3$ formation and removal processes in the different regions, although we find that the dominant variables such as temperature and photolysis rates remain important for both regions.*

*We also calculate the feature importance at the TOAR measurement locations only, and compare use of surface $O_3$ from TOAR and CAMS separately to explore the sensitivity of our results to the choice of reference data. We find that the feature importance at these locations differs markedly from that over the globe for some variables, particularly for OH and for geographical variables such as latitude and longitude. These differences reflect the limited spatial coverage of measurement sites and the narrower range of chemical environments sampled. However, the feature importance is very similar whether using TOAR measurements or CAMS reanalysis $O_3$ at these same locations, demonstrating that these datasets provide very similar information, and this lends confidence in our choice of CAMS reanalysis data in our analysis.*

[Figure]

**Figure 5.** *Importance of different variables to surface O$_3$ biases calculated by the Shapley additive explanations framework (SHAP) for the deep learning model. **(a)** Feature importance over the globe derived from CAMS reanalysis data. Strong positive (r > 0.7) and negative (r < -0.7) relationships between O$_3$ biases and variable values are shown in red and blue, respectively, while weaker relationships are shown in grey. **(b)** Feature importance over land and ocean regions derived from CAMS reanalysis data. **(c)** Comparison of feature importance inferred from CAMS and TOAR separately at the TOAR measurement locations only. Error bars show one standard deviation of feature importance in % for each variable.*

Page 17, line 360:

*... and will explore the challenges of data sparsity and spatial representativeness associated with use of surface measurements directly in future work. The development of a robust and reliable surface O$_3$ climatology based on observations would be particularly useful to improve assessment of model biases.*

3. **The conclusion from the SHAP based ranking of feature importance remains too general and I am not sure what can be learned from that. All the mentioned factors contribute to ozone variability, and it is likely that that is also reflected in their correlation with the inter-model differences. The often-mentioned biases of the emissions were seemingly not identified as a large contributor to ozone biases, which is surprising as the relatively small impact of the NO concentrations. I also wonder why ozone concentrations itself were not used as an input variable for the training. I am not saying the results are wrong, I would only suggest avoiding the over-interpretation of these correlations into actual causes of the biases. But, if the authors can demonstrate that so far unknown deficiencies of UKSEM1 were identified and potentially even fixed based on the finding of the feature importance, this should be mentioned more convincingly.**

The principal goals of our study are to demonstrate the new approach to bias correction and show how important this can be for our assessment of future changes in surface ozone with current models. Extracting the feature importance from the bias correction approach applied provides additional useful information, but does not constitute a vital result upon which our conclusions depend. We

have tried to be clear that the importance of features should not be interpreted as causation; there is merely an association of the biases with these features. It is not the purpose of this study to identify specific process weaknesses in the model. Despite this, and in contrast to the reviewers assertion, additional useful insight can still be gained from this information. The importance of temperature is clear, and suggests weakness in chemistry or boundary layer processes. As the reviewer notes, NO is not a particularly important feature, suggesting that $NO_x$ emissions are not a major source of biases at this scale, but we note that VOC are substantially more important, highlighting that the combination of chemistry and emissions is important. We also note that NO is more important for biases over the ocean than over the land in our new text and figure 5b, and we have now added this analysis in the manuscript to emphasise that there are substantial regional variations reflecting different chemical and physical environments.

We explored the use of ozone as an input variable to the machine learning model (along with many other variables), but found that it was not necessary to achieve good performance. The relationship between ozone biases and ozone is of course strong, but we did not feel that this provided any useful additional information about why biases occur.

To address the reviewer's concerns, we have reiterated the importance of not overinterpreting the feature importance, and we have adjusted out statements about what can be learned from these (in response to point 2 above), and discussed the underlying weaknesses in UKESM1 suggested by other studies in the section on feature importance.

Page 11, line 243:
*... or to other processes for which temperature is a proxy, and this explains the seasonality of the reversal in $O_3$ biases from winter to summer in the Northern hemisphere.*

*Specifically for UKESM1, Archibald et al. (2020) found that the $O_3$ responses to same temperature changes in two chemical mechanisms (including StratTrop of UKESM1) are distinct, suggesting that temperature may be a main source of biases. In addition, more comprehensive chemistry schemes based on StratTrop further enlarge the $O_3$ biases in summer, reported by Archer-Nicholls et al. (2021) and Liu et al. (2022), indicating that the chemistry scheme itself may not be the main cause of biases but the external variables driving the scheme e.g. temperature and photolysis rates may be more important. We note that the relationships derived between the variables and $O_3$ biases reflect association, not causation, and that specific processes cannot be identified directly as the sources of biases. However, the association revealed provides some hints for the underlying processes associated with relevant variables.*

**References:**

Archer-Nicholls, S., Abraham, N. L., Shin, Y., Weber, J., Russo, M. R., Lowe, D., Utembe, S., O'Connor, F., Kerridge, B., and Latter, B.: The Common Representative Intermediates Mechanism version 2 in the United Kingdom Chemistry and Aerosols Model, Journal of Advances in Modeling Earth Systems, 13, e2020MS002420, 2021.

Archibald, A. T., Turnock, S. T., Griffiths, P. T., Cox, T., Derwent, R. G., Knote, C., and Shin, M.: On the changes in surface ozone over the twenty-first century: sensitivity to changes in surface temperature and chemical mechanisms, Philosophical Transactions of the Royal Society a-Mathematical Physical and Engineering Sciences, 378, 20190329, 2020.

Betancourt, C., Stomberg, T. T., Edrich, A. K., Patnala, A., Schultz, M. G., Roscher, R., Kowalski, J., and Stadtler, S.: Global, high-resolution mapping of tropospheric ozone – explainable machine learning and impact of uncertainties, Geosci. Model Dev., 15, 4331-4354, 10.5194/gmd-15-4331-2022, 2022.

Chang, K. L., Cooper, O. R., West, J. J., Serre, M. L., Schultz, M. G., Lin, M., Marécal, V., Josse, B., Deushi, M., Sudo, K., Liu, J., and Keller, C. A.: A new method (M3Fusion v1) for combining observations and multiple model output for an improved estimate of the global surface ozone distribution, Geosci. Model Dev., 12, 955-978, 10.5194/gmd-12-955-2019, 2019.

Huijnen, V., Miyazaki, K., Flemming, J., Inness, A., Sekiya, T., and Schultz, M. G.: An intercomparison of tropospheric ozone reanalysis products from CAMS, CAMS interim, TCR-1, and TCR-2, Geosci. Model Dev., 13, 1513-1544, 10.5194/gmd-13-1513-2020, 2020.

Ivatt, P. D., and Evans, M. J.: Improving the prediction of an atmospheric chemistry transport model using gradient-boosted regression trees, Atmos. Chem. Phys., 20, 8063-8082, 10.5194/acp-20-8063-2020, 2020.

Keller, C. A., Evans, M. J., Knowland, K. E., Hasenkopf, C. A., Modekurty, S., Lucchesi, R. A., Oda, T., Franca, B. B., Mandarino, F. C., Díaz Suárez, M. V., Ryan, R. G., Fakes, L. H., and Pawson, S.: Global impact of COVID-19 restrictions on the surface concentrations of nitrogen dioxide and ozone, Atmos. Chem. Phys., 21, 3555-3592, 10.5194/acp-21-3555-2021, 2021.

Liu, Z., Doherty, R. M., Wild, O., O'Connor, F. M., and Turnock, S. T.: Tropospheric ozone changes and ozone sensitivity from the present day to the future under shared socio-economic pathways, Atmos. Chem. Phys., 22, 1209-1227, 2022.

Schultz, M. G., Schröder, S., Lyapina, O., Cooper, O. R., Galbally, I., Petropavlovskikh, I., Von Schneidemesser, E., Tanimoto, H., Elshorbany, Y., and Naja, M.: Tropospheric Ozone Assessment Report: Database and metrics data of global surface ozone observations, Elementa: Science of the Anthropocene, 5, 2017.

---

## Author Response (AR2)

Dear editor:

We thank you for your comments to further improve our manuscript, and address your concerns below. For clarity, the comments are given in bold, followed by our responses. The modified text in our revised manuscript is given in italics and blue.

**I will accept the manuscript subject to minor revisions. Reviewer 2 suggested a rejection of this manuscript, as using the CAMS model as a reference case was subject to substantial biases (essentially the model is not strongly constrained by observations at the surface). I do not concur with the reviewer 2's recommendation, and motivate this as follows.**

**In your revised manuscript you convincingly demonstrate there is nevertheless a substantial merit in using CAMS as the observational 'truth'. This is because the biases in the UKESM1 are substantially higher than in CAMS.**

We thank you for your recognition of the merit of our work and the revised version of the manuscript.

**However, it would be good to highlight in abstract, and conclusions that there is also a risk of over-interpretation when using another model for the machine learning based bias correction, when essentially similar processes may have lead to biases in the models, output fields are more smooth than in the real-world etc.**

We thank you for pointing out that using non-observation based data as reference may still lead to biases in explaining deep learning outputs. We did this because of its relatively small biases in surface $O_3$, as you mentioned above, which indeed provides a great opportunity to correct biases for other models with large errors such as UKESM1. However, we acknowledge that the systematic errors due to the issues in processes and spatial representativeness may occur in all physical models, and cannot be fully eliminated, which may impact the bias correction. We have now acknowledged this point in both the abstract and conclusions.

Page 1, line 9:
*... Atmospheric chemical species such as the hydroxyl radical, nitric acid and peroxyacyl nitrate show strong positive relationships with ozone biases on a regional scale. These relationships reveal the conditions under which ozone biases occur, although they reflect association rather than direct causation.*

Page 17, line 361:
*... We also note that there are weaknesses in the representation of $O_3$ in the reanalysis data which are likely to affect the magnitude of the biases we have derived. However, we have successfully demonstrated the feasibility of bias correction using this data, and will explore the challenges of data sparsity and spatial representativeness associated with use of surface measurements directly in future work.*

**The conclusions at line 333-336 already gives some hints on how you want to address these issues, but need to be expanded to better indicate what the limitations of the study are. Also please discuss to what extend the method is applicable for other models which are less biased (i.e. similar magnitude of biases as CAMS).**

We thank you for the suggestions. Explaining machine learning outputs is always difficult because it is based on statistics, and that is where the limitation of the study is. It is difficult to use a few variables to identify the specific processes that have the largest biases, and it is also a challenge. We have acknowledged the limitations in the Conclusions section, and discussed the method that may be more suitable for models with small biases.

Page 17, line 359:

*... However, we demonstrate that the relationships between the variables with the highest feature importance and surface O₃ biases are intuitive, e.g. with temperature and photolysis rates, and this provides useful insight for further model improvement. While we are not able to identify the specific processes leading to biases using this approach, it allows us to target processes that are most sensitive to these variables. It would be valuable to develop explainable machine learning algorithms to use for bias correction.*

Page 17, line 364:

*... However, we have successfully demonstrated the feasibility of bias correction using this data, and will explore the challenges of data sparsity and spatial representativeness associated with use of surface measurements directly in future work. This approach should also be directly applicable for models with smaller initial biases, and in this case it would be particularly valuable to consider daily or hourly mean O₃ to explore representation of synoptic and diurnal variations in O₃. However, the development of a robust and reliable surface O₃ climatology based on observations would be particularly useful to improve assessment of model biases.*